# Assisted reproductive technologies are associated with limited epigenetic variation at birth that largely resolves by adulthood

Boris Novakovic[1,2], Sharon Lewis[1,2], Jane Halliday[1,2], Joanne Kennedy[1], David P. Burgner [1,2,3,4], Anna Czajko[1], Bowon Kim[1], Alexandra Sexton-Oates[1], Markus Juonala [1,5,6], Karin Hammarberg [7,8], David J. Amor[1,2,4], Lex W. Doyle [1,2,9,10], Sarath Ranganathan[1,2,4], Liam Welsh[1,4], Michael Cheung[1,2,4], John McBain[11], Robert McLachlan[12,13,14] & Richard Saffery [1,2]

More than 7 million individuals have been conceived by Assisted Reproductive Technologies (ART) and there is clear evidence that ART is associated with a range of adverse early life outcomes, including rare imprinting disorders. The periconception period and early embryogenesis are associated with widespread epigenetic remodeling, which can be influenced by ART, with effects on the developmental trajectory in utero, and potentially on health throughout life. Here we profile genome-wide DNA methylation in blood collected in the newborn period and in adulthood (age 22–35 years) from a unique longitudinal cohort of ART-conceived individuals, previously shown to have no differences in health outcomes in early adulthood compared with non-ART-conceived individuals. We show evidence for specific ART-associated variation in methylation around birth, most of which occurred independently of embryo culturing. Importantly, ART-associated epigenetic variation at birth largely resolves by adulthood with no direct evidence that it impacts on development and health.

[1] Murdoch Children's Research Institute, Parkville, VIC 3010, Australia. [2] Department of Paediatrics, University of Melbourne, Parkville, VIC 3010, Australia. [3] Department of Paediatrics, Monash University, Clayton, VIC 3800, Australia. [4] The Royal Children's Hospital, Parkville, VIC 3052, Australia. [5] Department of Internal Medicine, University of Turku, 20500 Turku, Finland. [6] Division of Medicine Turku University Hospital, 20500 Turku, Finland. [7] Global Public Health, Public Health and Preventive Medicine, Monash University, Melbourne, VIC 3800, Australia. [8] Victorian Assisted Reproductive Treatment Authority, Melbourne, VIC 3000, Australia. [9] Department of Obstetrics and Gynaecology, University of Melbourne, Melbourne, VIC 3010, Australia. [10] Research Office, The Royal Women's Hospital, Parkville, VIC 3052, Australia. [11] Reproductive Services, Royal Women's Hospital, Parkville, VIC 3052, Australia. [12] Hudson Institute of Medical Research, Clayton, VIC 3168, Australia. [13] Monash IVF Group Pty Ltd, Richmond, VIC 3121, Australia. [14] Department of Obstetrics and Gynaecology, Monash University, Clayton, VIC 3800, Australia. Correspondence and requests for materials should be addressed to R.S. (email: richard.saffery@mcri.edu.au)

A ssisted Reproductive Technologies (ART) have resulted in more than 7 million births since 1978[1]. Today, ART procedures are diverse, spanning the relatively less invasive intervention of gamete intra-fallopian transfer (GIFT) and intra-uterine insemination (IUI)[2], through fertilization of gametes in vitro with culturing (in vitro fertilization, IVF), to the more recent direct injection of a sperm into an ovum (intracytoplasmic sperm injection, ICSI), followed by culturing, with or without subsequent embryo freeze/thawing[3,4].

Mounting evidence suggests that early periconceptional exposures (such as ART) may influence developmental trajectories in offspring[5,6]. ART conception is associated with an approximately two-fold increased risk of preterm birth, low birth weight, being small for gestational age or perinatal mortality[7–9]. However, despite the continuing expansion of ART worldwide[10–12], few studies have investigated the potential long-term health outcomes associated with assisted conception, or the potential underlying molecular and cellular variations. Some but not all studies of children and adolescents born following ART report possible increased cardiovascular[13,14] and metabolic risks[13,15], raised plasma lipids, and higher rates of obesity[15]. Large epidemiological studies also suggest an increased risk of rare imprinting disorders in association with epigenetic disruption early in development[16]. Notwithstanding, meta-analyses and systematic reviews suggests a dearth of compelling data supporting any consistent ART-associated adverse outcomes in either children or adults[17–25].

The periconceptional period is associated with widespread epigenetic remodeling in gametes and the early embryo[26]. It is therefore plausible that the early epigenetic profile is influenced by ART processes, with potential to alter the developmental trajectory in utero and throughout life[27,28]. For example, the hormonal milieu created by ovarian stimulation and the in vitro culturing of the embryo have both been suggested as processes that can alter epigenetic profile in ART progeny[29,30], however published data are circumstantial, limited, and at times contradictory[31]. A recent review summarizes the potential adverse effects on long-term health associated with ART, some of which may be attributable to epigenetic variation induced in the periconceptional period[6]. Further evidence suggests that variation in the developing epigenetic profile may occur at repetitive elements that make up a large proportion of the human genome[32].

Given the rising rates of ART pregnancies internationally[4], limited evidence of potential adverse short to medium term health outcomes, and the relatively limited number of studies of epigenetic variation in association with ART, it is imperative that any underlying epigenetic variation induced by ART is fully explored in humans, particularly as this population ages. This is especially important given emerging links between epigenetic variation and a range of adverse adult-onset cardiometabolic, neurodevelopmental, and respiratory conditions[33,34].

We previously established a cohort of singleton ART-conceived young adults (aged 18–28 years) and a matched non-ART group from the same source population, and using a telephone interview found an increased rate of maternally reported hospital admissions, atopic respiratory conditions, and metabolic/endocrine/nutritional disease (ICD-10 coding category) in the ART-conceived group[35]. More recently, we assessed vascular, cardiometabolic, anthropometric, and respiratory health clinically in a subset of the original cohort, now aged 22–35 years, and found no evidence of adverse health outcomes associated with ART conception[36]. In the current study, we perform a longitudinal Epigenome-wide Association Study (EWAS) of these ART and non-ART-conceived individuals from the neonatal period through to adulthood, spanning up to 35 years since birth.

## Results

**ART-associated differential methylation at birth is largely attenuated in adulthood.** To investigate whether DNA methylation levels in blood differ between ART-conceived individuals relative to non-ART conceived individuals, we analyzed epigenome-wide methylation data in neonatal (Guthrie spot) and adult peripheral whole blood using the EPIC array. DNA methylation status was generated for 149 neonatal (84♀ 65♂) and 158 adult (87♀ 71♂) ART-conceived individuals and for 58 neonatal (37♀, 21♂) and 75 adult (51♀, 24♂) non-ART conceived individuals (Fig. 1a).

In neonatal blood, we identified 2340 (out of total 724,897 probes) differentially methylated probes (DMPs) between ART and non-ART groups following FDR correction for multiple testing, but none in the adult samples (out of total 766,247 probes) (Fig. 1b). The mean (SD) methylation difference ($\Delta\beta$) of the 2340 DMPs between the groups was $0.026 \pm 0.013$ (largest effect of 0.129 (i.e., 12.9%)). The majority of DMPs (79.1%), showed a higher DNA methylation level among ART offspring in neonatal blood compared with non-ART offspring. Restricting DMPs to those that showed greater than 5% difference between groups ($-0.05 \geq \Delta\beta \geq 0.05$) revealed 116 DMPs (85%) with higher, and 20 (15%) with lower methylation in the ART group relative to the non-ART group in neonatal blood (Fig. 1b, Supplementary Data 1). Despite not reaching significance following FDR correction in adulthood, six of these 136 DMPs were also differentially methylated by ≥5% in adulthood (Fig. 1c), albeit attenuated in magnitude (Fig. 1d, Supplementary Data 2). We did not observe any anti-correlating probes (e.g., hypomethylated in ART neonates but hypermethylated in ART adults). Of the 136 DMPs showing greater than 5% difference between groups, all but one were within 1 Mb of a gene transcription start site, with 4 genes having 2 DMPs in their vicinity (Supplementary Data 1).

Next we examined differentially methylated regions (DMRs), which contain multiple DMPs that show correlative methylation. DMRs, defined as a region containing ≥3 DMPs, at least one of which have a $\Delta\beta \geq 5\%$, were identified using DMRcate (Fig. 2a). In total 18 DMRs (comprising 106 total probes) were identified in neonatal blood (Fig. 2b) and 4 DMRs (comprising 27 probes) were identified in adulthood (Fig. 2c). Three DMRs common to both time-points were found near the genes *CHRNE* (7 probes), *PRSS16* (3 probes), and *TMEM18* (9 probes), with the same direction and similar level of DNA methylation change (Fig. 2d, e). The full list of significant DMPs in neonatal blood, the highest ranked in adult blood, and the DMRs at both time-points, are listed in Supplementary Data 1–4.

**CHRNE exhibits both age-specific and ART-specific differential methylation.** In order to further explore the specificity of the observed ART-associated differential methylation, we examined methylation profiles more broadly around the identified DMRs of interest, at both time-points. Probes around all three DMRs identified at birth showed a complex pattern of ART-associated and age-associated differential methylation, with little difference between the ART and non-ART groups at probes outside the identified DMRs (Fig. 3, Supplementary Figs. 2 and 3).

Initial identification of DMPs associated with ART revealed two probes in close proximity to each other within the *CHRNE* gene, displaying a loss of methylation at genome-wide significance at birth (cg10553748, $\Delta\beta$ −0.063; cg24768135, $\Delta\beta$ −0.11), (Fig. 3a(i)). The same DMPs showed a slightly reduced methylation difference in adulthood ($\Delta\beta$ −0.053 and −0.062, respectively) and did not reach FDR significance (Fig. 3a(ii)). DMR analysis revealed a regional loss of methylation in

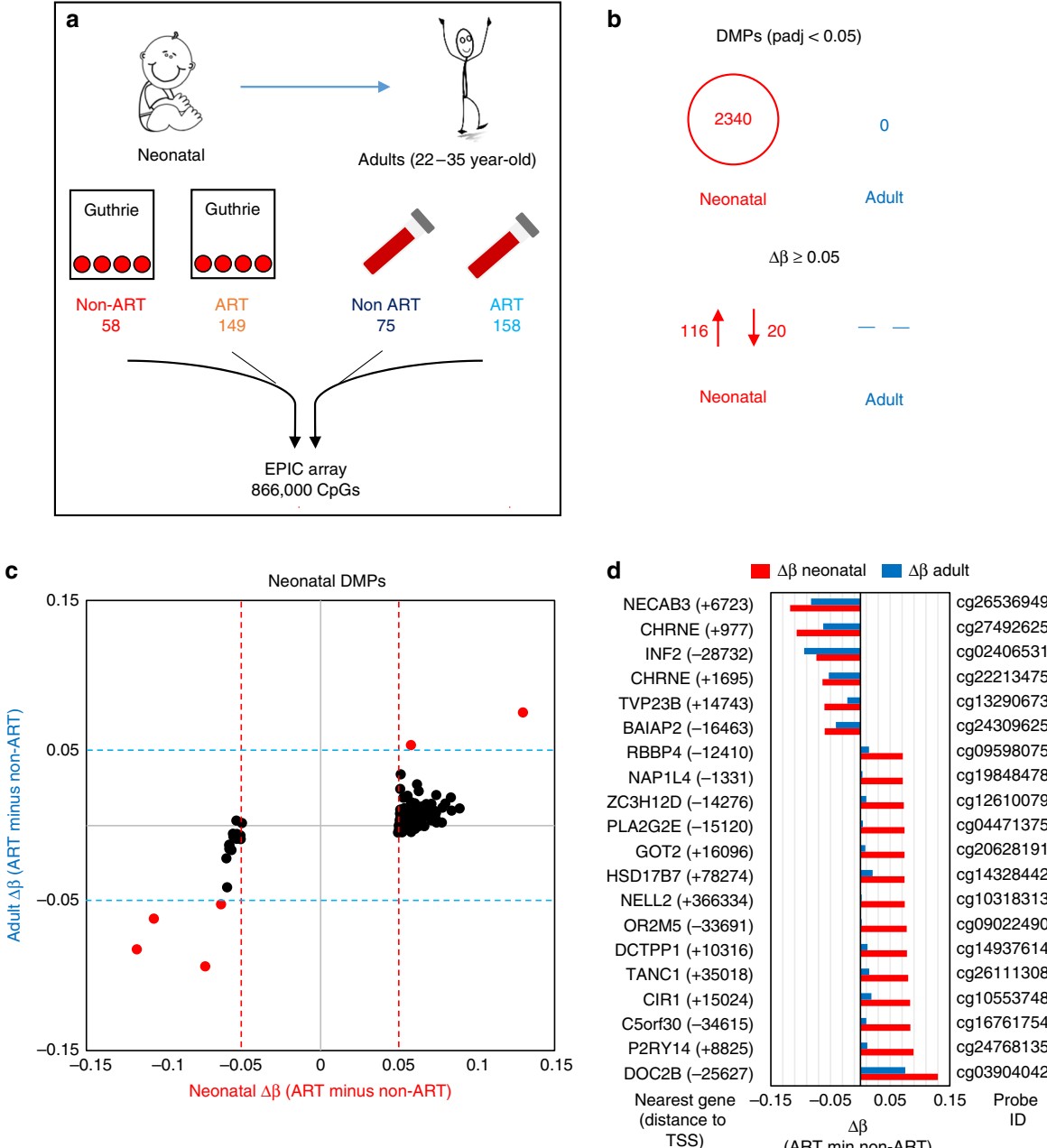

**Fig. 1** Study design and identification of ART-associated differentially methylated probes in neonatal and adult blood. **a** Summary of the longitudinal EWAS. **b** Number of DMPs that pass an adjusted *p*-value cut-off of <0.05 and a Δβ cut-off of ≥0.05. **c** Correlation plot of mean Δβ between ART and non-ART groups in neonatal (*x* axis) and adult (*y* axis) blood at neonatal DMPs. Δβ is always calculated as mean ART DNA methylation minus mean non-ART DNA methylation. Red dots represent probes that show a Δβ ≥ 5% between ART and controls in both neonatal and adult blood (red dotted line and blue dotted line designate the 0.05 mark in all directions), black dots represent probes that only fulfill DMP criteria at birth. **d**. Bar plot of neonatal (red) and adult (blue) blood Δβ values for top ranked probes based on methylation change in neonatal blood, with accompanying probe ID, name of nearest gene and distance to gene TSS in brackets. While most probes no longer show differences in adult blood (at adjusted *p*-value <0.05 (Bayesian levene's test)), there are several probes that show a persistent change in methylation. *n* = 207 biologically independent birth samples, *n* = 233 biologically independent adult samples

association with ART, encompassing 7 probes in total (DMR1) and spanning approximately 1.89 kb. Average loss of methylation in the ART group at probes within this DMR was Δβ −0.053 and −0.056 in neonatal and adult blood, respectively.

Evidence of differential methylation in association with increasing age was also apparent within *CHRNE*, both at the DMP and DMR level (Fig. 3b, c). A sub-region of the ART-associated DMR1 (DMR1a), showed evidence of higher methylation in adulthood relative to infancy (Δβ = 0.108, $p = 3.4 \times 10^{-24}$

(Bayesian levene's test) for non-ART neonatal vs. non-ART adult), with a reduction of methylation in ART relative to non-ART at both ages ($p = 0.003$ and $p = 2.47 \times 10^{-10}$ (Bayesian levene's test)), whereas a second region of the same DMR (DMR1b) was differentially methylated specifically in association with ART. In addition, within *CHRNE*, an age-specific (aDMR, DMR2) encompassing 5 probes, and a single aDMP (cg20814095), showed an-age specific methylation difference not sensitive to ART at either age (Fig. 3c).

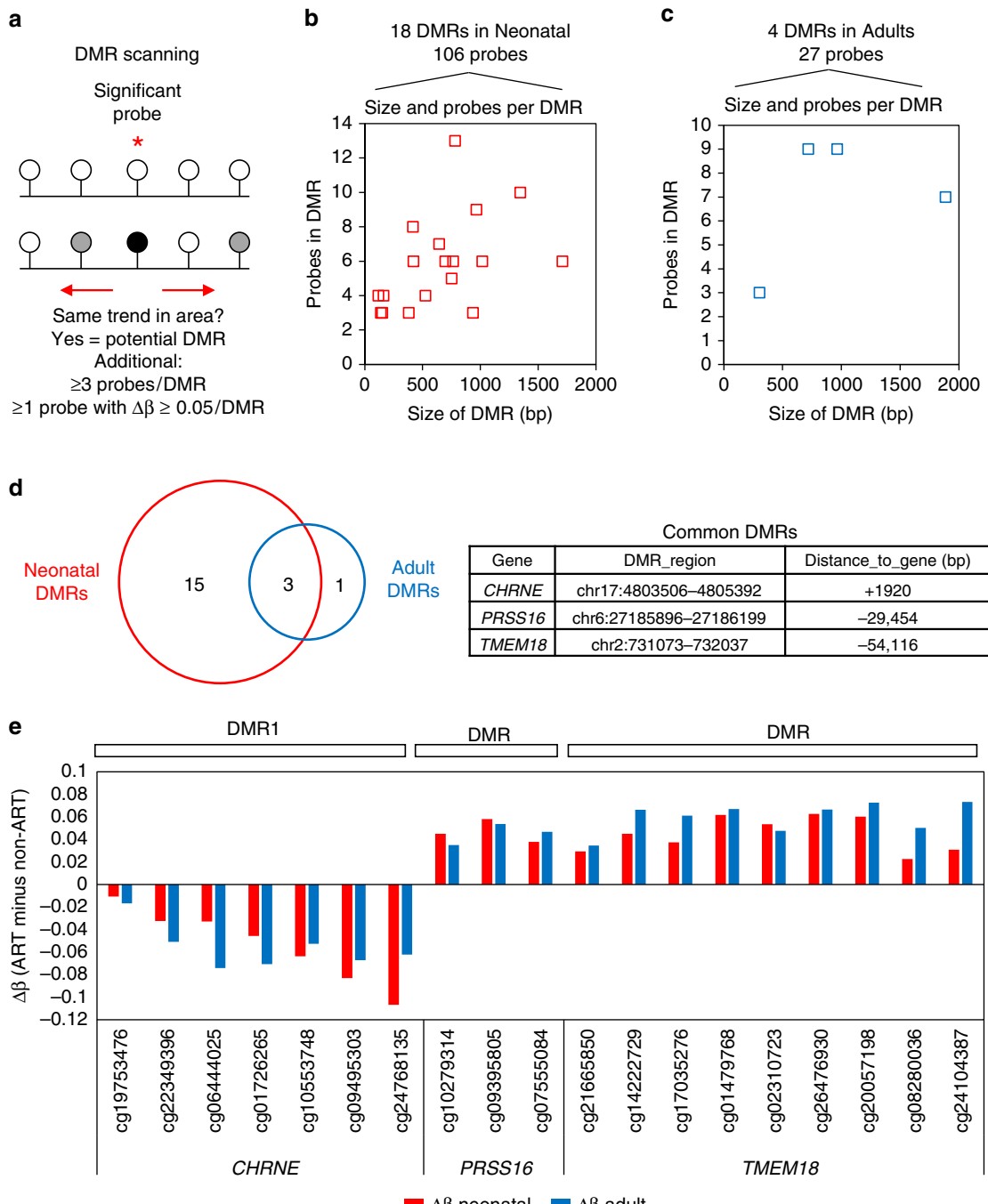

**Fig. 2** Identification of ART-associated differentially methylated regions in neonatal and adult blood. **a**. Overview of the strategy used to identify DMRs. For each probe with an adjusted $p$-value <0.05 (Bayesian levene's test) identified during linear regression analysis, DMRcate was used to scan the surrounding region for probes that show the same general DNA methylation change. Once regions were identified by DMRcate, the following cut-offs were used: at least 3 probes in the region, at least 1 of which has a $\Delta\beta \geq 5\%$. **b**. Scatterplot of DMR size and number of probes within a DMR in neonatal blood. A total of 106 probes within 18 DMRs were identified, with the size ranging from 100 to 1700 bp and number of probes per DMR ranging from 3 to 13. **c** Scatterplot of DMR size and number of probes within a DMR in adult blood. A total of 27 probes within 4 DMRs were identified, with the size ranging from 300 to 1900 bp and number of probes per DMR ranging from 3 to 9. **d** Venn diagram showing that three DMRs which overlap between neonatal and adult blood. The name of nearest gene, location of DMR, and distance to gene TSS is shown for the common DMRs. **e** Column graph showing $\Delta\beta$ between mean ART and mean control for individual probes within the three common DMRs for neonatal (red) and adult (blue) blood. Probe ID and gene name are shown on the $x$ axis

**Independent replication of ART-associated differential methylation in infancy.** In order to test our ART-associated specific differential methylation in an unrelated cohort, we analyzed a dataset previously published by Estill et al. that was acquired using the Illumina Infinium HumanMethylation450K array (from now on referred to as the '450K dataset') (GSE79257) generated from 94 ART and 43 non-ART neonatal blood spots[37]. The disadvantage of using the 450K array for validation is that it does not include all EPIC probes and therefore generates a lower resolution picture of DNA

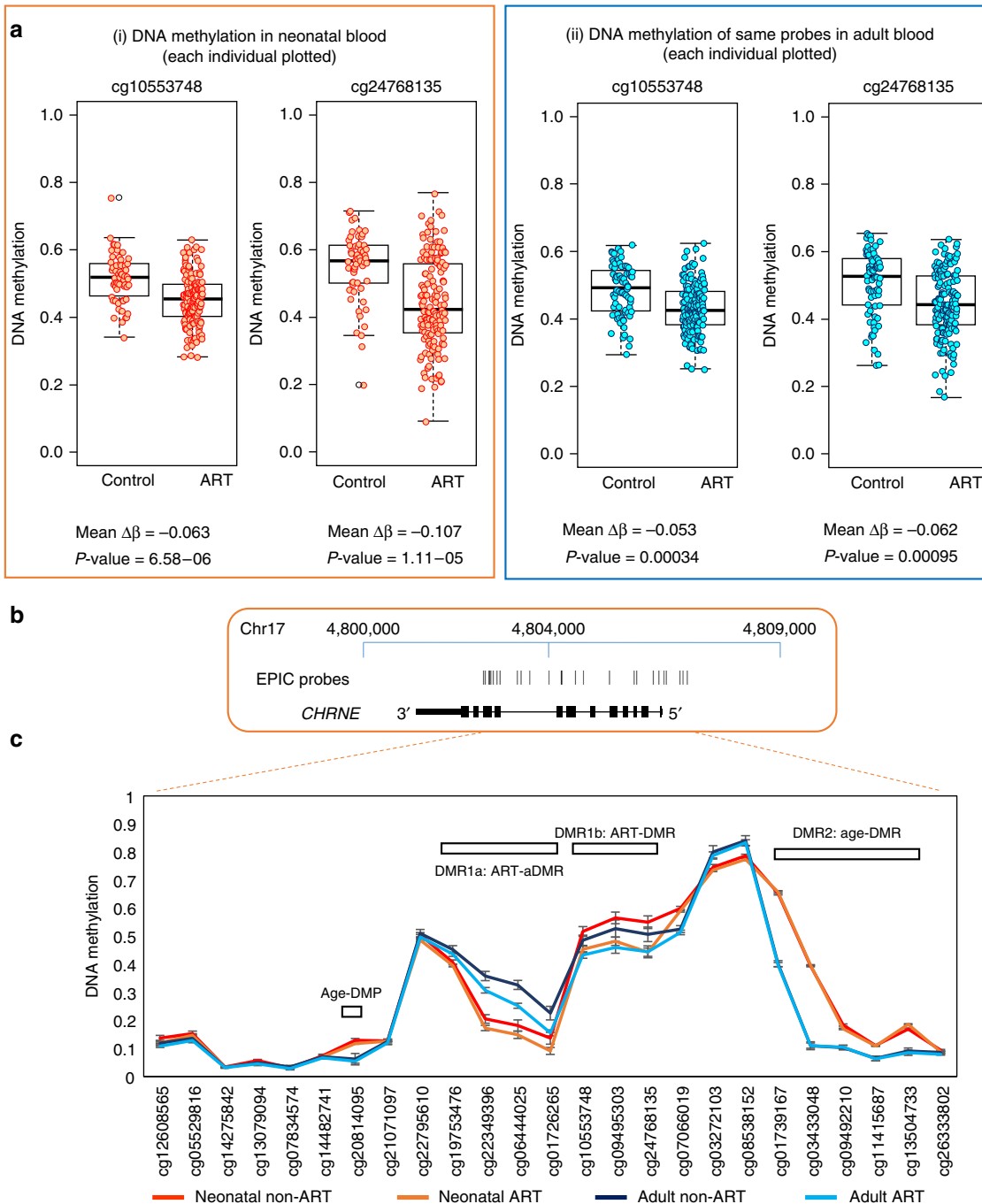

**Fig. 3** Detailed DNA methylation map of the *CHRNE* gene. **a** (i) Boxplot and dot-plot of DNA methylation for individual neonatal control and ART samples at the two probes within the *CHRNE* gene that showed a significant difference between groups (adjusted *p*-value <0.05 (Bayesian levene's test)). $n = 207$ biologically independent birth samples, $n = 233$ biologically independent adult samples. (ii) Boxplot and dot-plot of the same two probes in individual adult control and ART samples. The change is no longer significant in adult samples after correction for multiple testing, but the direction of methylation change persists. Boxplot elements are: center line-median; box limits-upper (Q3) and lower (Q1) quartiles; whiskers–smallest and largest non-outlier; points-outliers. **b**. Map of the *CHRNE* gene in hg19, showing EPIC probe locations. **c** Mean DNA methylation level at *CHRNE* for neonatal and adult non-ART and ART groups. Error bars are 95% confidence intervals. DMR1 is split into two: DMR1a that shows both ART and age specific DNA methylation differences and DMR1b that only shows ART-specific DNA methylation change. In addition, an age-specific DMR (DMR2) and a DMP are highlighted

methylation profile. Despite this, there is a strong correlation between methylation values generated by the 450K and EPIC arrays, which allows us to confirm a subset of our findings[38]. Of the 136 ART-associated DMPs we identified in neonatal blood on the EPIC array, data for 50 probes were also present in the 450K dataset, of which 14 (28%) showed evidence of

differential methylation in association with ART (6 of 50 probes at *p*-value < 0.05 and a further 8 at *p*-value < 0.10 (Bayesian levene's test); Supplementary Data 5). Further examination of methylation of probes from the 3 strongest DMRs in our dataset (Fig. 4a), also covered by probes in the 450K dataset, revealed a replication of the *CHRNE* and *PRSS16*

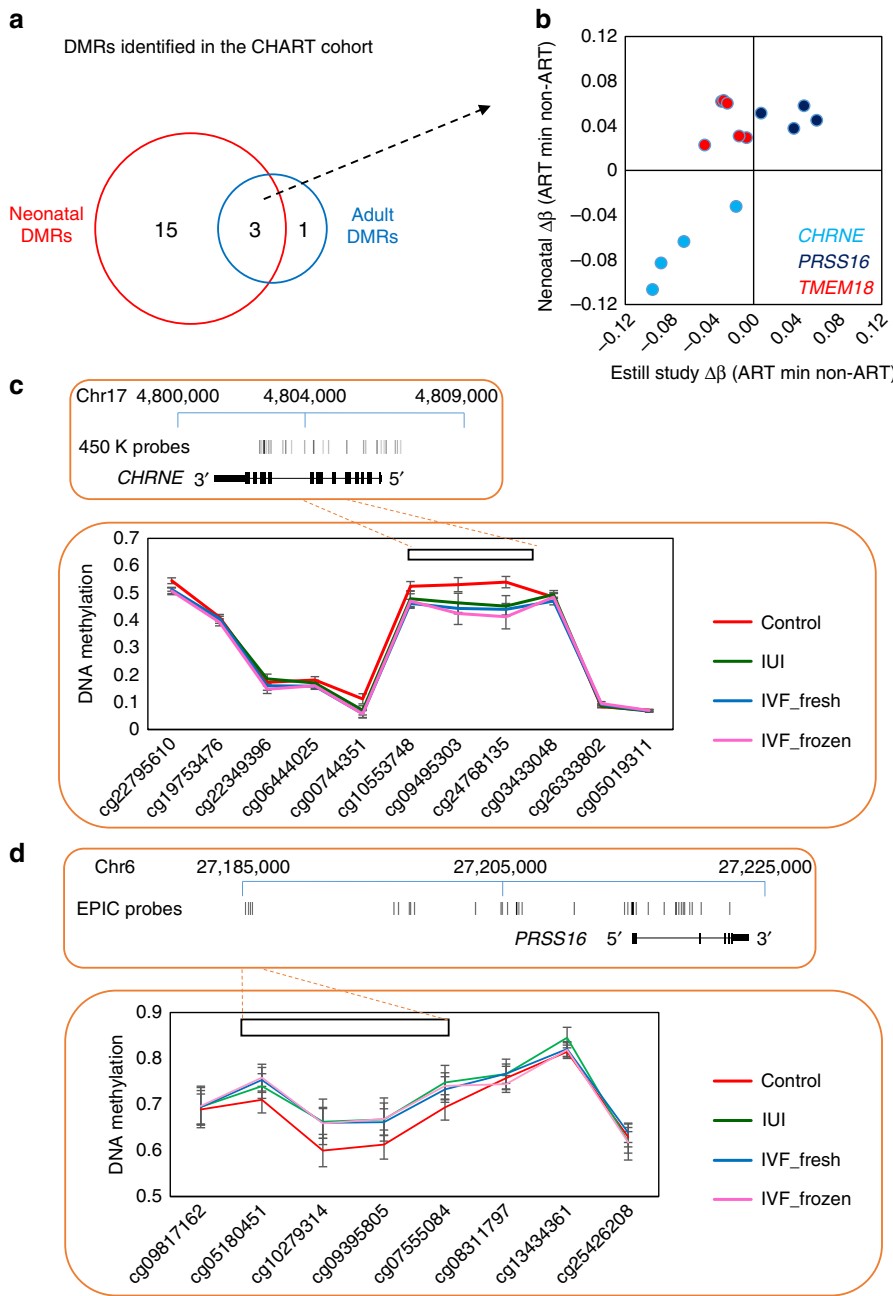

**Fig. 4** Validation of ART-associated differentially methylated regions in an unrelated cohort. **a**. Summary of three ART-DMRs that are detected in both neonatal and adult blood. **b** Scatterplot of individual probes within the three DMRs at *CHRNE*, *PRSS16* and *TMEM18* genes, with mean Δβ for neonatal blood in our cohort (CHART) shown on *y* axis and mean Δβ for neonatal blood in the 450K cohort shown on *x* axis. DMRs at *CHRNE* and *PRSS16* correlate well between the two studies. **c** Map of the *CHRNE* gene in hg19, showing EPIC probe locations and mean DNA methylation level at the ART-DMR in the 450K study. **d** Map of the *PRSS16* gene in hg19, showing EPIC probe locations and mean DNA methylation level at the ART-DMR in the 4 study. Error bars are 95% CI. *n* = 133 biologically independent birth samples

ART-associated DMRs, both in terms of direction of effect and magnitude of difference, with little supporting data obtained for *TMEM18* (Fig. 4b, Supplementary Data 6).

The samples used in the 450K dataset were separated into three groups: IVF with fresh embryo, IVF with thawed frozen embryo, and intra uterine insemination (IUI) (i.e., no IVF or embryo culturing)[37]. A change in methylation at the *CHRNE* and *PRSS16* DMRs was observed in all ART sub-types (Fig. 4c, d), suggesting that ART-associated effects are not associated with embryo culture specifically, but with other steps in the ART process, or potentially infertility in general. Conversely, an exploration of the top

differentially methylated region in the 450K dataset, (*SPATC1L*) in our dataset revealed some supporting evidence of an ART-associated DMR. Previous analysis identified two DMRs in *SPATC1L*, one at the promoter (7 probes with mean Δβ 0.088) and one in the gene body (6 probes with mean Δβ −0.14) (Supplementary Fig. 4A). We only observed a difference between the non-ART and ART groups in our data at the promoter DMR, with a mean change in DNA methylation of −0.03 Δβ (Supplementary Fig. 4B), which is the same direction, but significantly smaller, compared with the change previously reported.

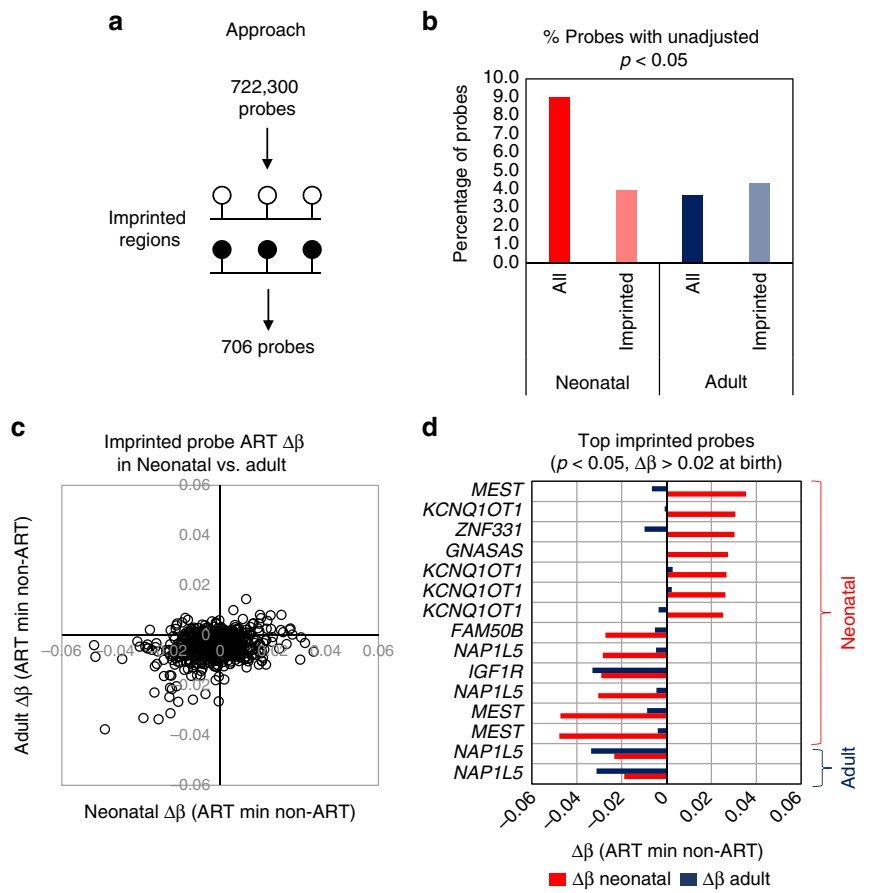

**Fig. 5** No evidence for ART-associated DNA methylation change at imprinted genes. **a** Strategy for analysis of DNA methylation at imprinted gene regions. A total of 706 probes mapped within 50 imprinted regions. **b** Column plot showing the percentage of all probes and probes at imprinted regions showing a difference between ART and non-ART groups at an unadjusted $p$-value <0.05 (Bayesian levene's test). This analysis shows that imprinted regions are less likely to have an ART-associated DNA methylation change than the average gene region. **c** Correlation plot of mean DNA methylation change between ART and non-ART in neonatal ($x$-axis) and adult ($y$-axis) blood at all imprinted probes. This shows that no probes show a $\Delta\beta \geq 4\%$ in adults, while only two probes meets this criteria in neonatal blood. **d** Bar plot of neonatal (red) and adult (blue) blood $\Delta\beta$ values for top ranked imprinted probes based on methylation change in neonatal or adult blood, with name of nearest gene

**Subtypes of ART associate with specific differential methylation in neonates**. The identification of differential methylation in *CHRNE* in association with IUI, without embryo culture, in the 450K dataset prompted us to explore the potential for different stages of ART processes to induce differential methylation in our CHART cohort. We categorized ART into three groups (i) ovarian stimulation only (gamete intrafallopian transfer, GIFT, $n = 35$), (ii) IVF with fresh embryos ($n = 75$) and (iii) IVF with thawed frozen embryos ($n = 30$) (Supplementary Fig. 5A). Plotting of differential methylation between groups confirmed a loss of methylation in *CHRNE* in all ART groups, including GIFT (no embryo culturing) (Supplementary Fig. 5B). Overall, there was no evidence that a subgroup was associated with a larger change in DNA methylation, though frozen IVF had a slightly lower median change in methylation relative to non-ART individuals (Supplementary Fig. 5C). Finally, we were interested to specifically compare culture (all IVF $n = 105$) and no-culture conditions (GIFT), but found no significant differences between these two subgroups (Supplementary Fig. 5D).

**Imprinted regions show limited evidence of differential methylation in association with ART**. Previous studies have demonstrated a relationship between ART conception and rare imprinted disorders[39], associated with aberrant DNA methylation

(summarized in a systemic review and meta-analysis)[16]. In addition, several studies have reported locus specific variation in imprinting associated regions in various tissues in association with ART conception, including in placenta[40,41] and cord blood[40]. We carried out a focussed analysis of 706 EPIC array probes that are located within 50 DMRs previously identified as being associated with imprinting[42] in our longitudinal dataset (Fig. 5a; Supplementary Data 7). In our original EWAS, only 2 imprinting associated probes showed evidence of ART-associated differential methylation (at birth), with a $\Delta\beta$ of 0.025 (cg12054318; adjusted $p$-value = 0.037) and $\Delta\beta$ of 0.027 (cg26104781; adjusted $p$-value = 0.0483). Nevertheless, we examined whether there was any evidence for enrichment of imprinted regions within the larger set of probes showing unadjusted $p$-value <0.05 (Bayesian levene's test), relative to non-imprinted regions. Whereas approximately 9% of all 722,000 probes showed some evidence of differential methylation in neonatal blood using this relaxed threshold, only 4% of imprinting-associated probes fell into this category (Fig. 5b). There was similarly no evidence of enrichment for imprinted regions in the ART-associated differential methylation results in adult blood (Fig. 5b). Where differential methylation at imprinting-associated probes was observed between ART and non-ART groups, the magnitude of difference was very modest ($\Delta\beta$ under 5% in all instances; Fig. 5c). Nevertheless, several imprinted regions showed weak evidence of coordinated gain or loss of methylation in DMRs at birth in

association with ART, though in most instances this was not apparent in adulthood (Fig. 5d). For example, 4 DMPs associated with *KCNQ1QT1* imprinting showed higher methylation in ART vs. non-ART blood, specifically in the neonatal period (Fig. 5d). These probes are adjacent and are the only ones in the broader *KCNQ1QT1* locus to show ART-associated differential methylation (Supplementary Fig. 6). In contrast, a single DMP in the *IGF1R* and two in the *NAP1L5* locus showed slightly greater methylation differences in adult blood between ART and non-ART groups relative to neonatal blood (Fig. 5d).

**No evidence for repeat-based 'global' methylation change in association with ART.** One of the most common proxy approaches to assess global DNA methylation is to focus on highly repetitive Alu and LINE1 elements that comprise 11% and 17% of the human genome, respectively (Fig. 6a)[43]. Taking a

mean value of methylation of probes in these elements provides a summary measure of 'global' methylation. The EPIC array platform contains >23,000 probes with homology to Alu and >29,000 to LINE1[43]. When the combined mean of these probes was compared across ART and non-ART groups, there was clear evidence of a gain in methylation with age (Fig. 6b, d), consistent with previous findings in buccal cells from birth to 7 years of age[44] but no evidence of a significant effect of ART at either age, despite a slightly higher median methylation level in both elements in neonatal blood (Fig. 6b, d). To further explore the potential for large scale altered genomic methylation of small magnitude, we compared the genome-wide average methylation level (GWAM[45]) of all 722,000 probes between ART and non-ART offspring at both ages (Fig. 6c). Unlike for Alu and LINE1, there was weak evidence for an effect of ART conception affecting this measure in adults ($\Delta\beta = 0.003$, $p = 0.002$ (Student's *t*-test)) (Fig. 6c, d).

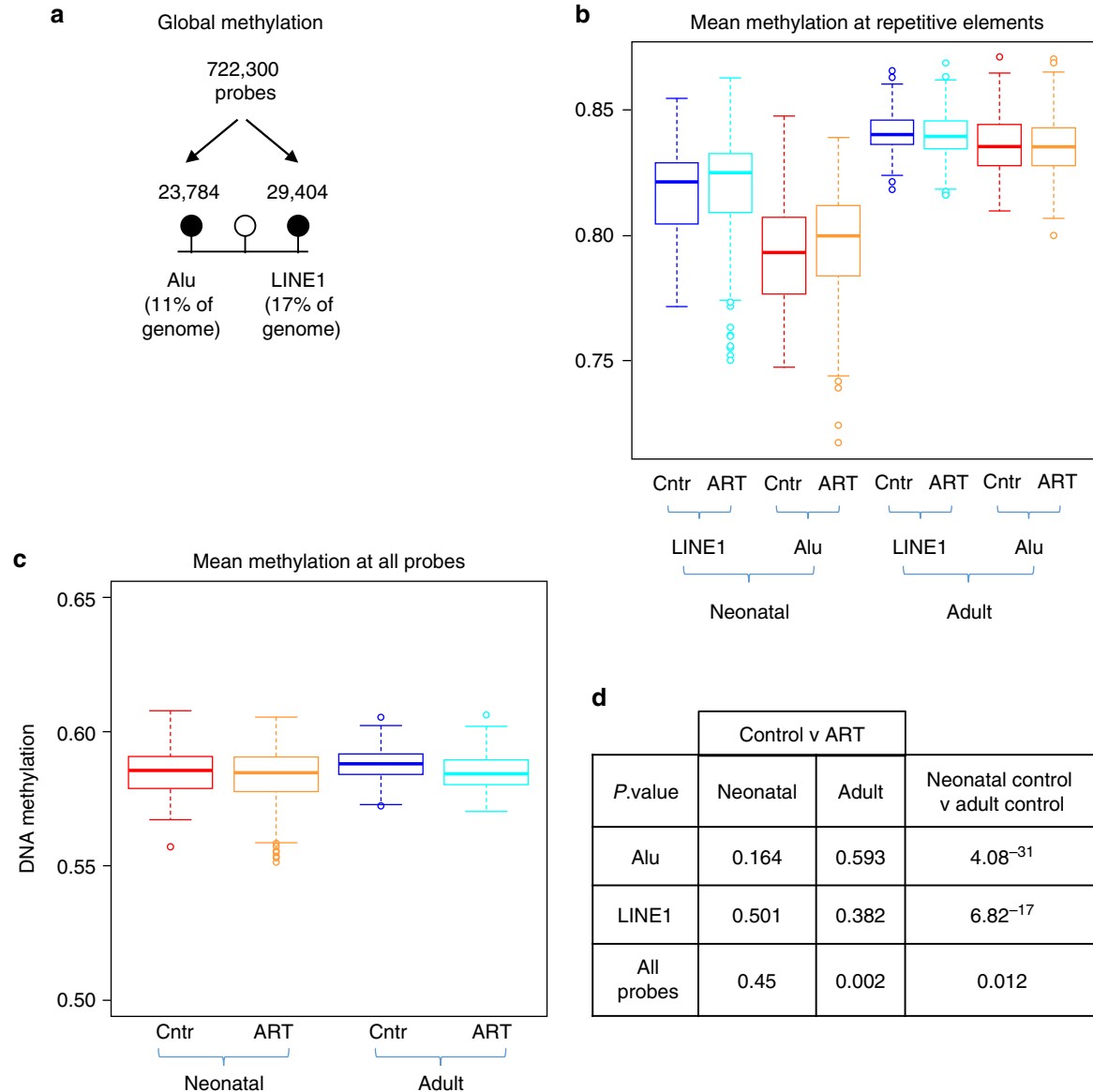

**Fig. 6** No evidence for ART-associated DNA methylation change at repetitive elements. **a** Global DNA methylation was assessed by grouping EPIC probes into Alu or LINE1 repetitive element regions using the REMP tool. **b** Boxplot of mean DNA methylation level of all probes at Alu and LINE1 in neonatal and adult blood. An age-effect is observed, but no significant differences between control and ART groups are detected. Boxplot elements are: center line-median; box limits-upper (Q3) and lower (Q1) quartiles; whiskers–smallest and largest non-outlier; points-outliers. **c**. Mean DNA methylation across all 722,301 probes in neonatal and adult blood. **d** P-values based on mean Alu, LINE1 and all probe DNA methylation for each group, using a Student's *t*-test

## Discussion

We performed a longitudinal analysis of DNA methylation profile in whole blood from early infancy to adulthood in a cohort of individuals conceived by ART and compared findings with non-ART conceived individuals. We found compelling evidence for specific ART-associated methylation variation around birth, some of which replicated in an independent cohort, with less evidence for persistence of differential methylation into adulthood. We found no evidence for an association between ART conception and DNA methylation at imprinting-associated regions, nor measures of global methylation relative to non-ART conception at either birth or adulthood. These findings demonstrate consistent ART-associated epigenetic variation by genome-wide analysis across two independent cohorts.

Given that the reported increase in rare imprinting disorders following ART conception is associated with variation in DNA methylation[16], it is logical that other genomic regions may also be sensitive to epigenetic disruption following ART. Several studies have directly tested this hypothesis using a combination of locus-fspecific, global and genome-wide approaches, with some finding no evidence of ART-associated epigenetic variation[40,46], while others report evidence of associations across different tissues and time-points (discussed below). Studies to date have generally been heterogeneous in design, have focussed on different tissues, ages, or genomic regions, used a variety of measurement approaches and have small sample sizes—all factors that likely contribute to the lack of replication of findings.

The effects of ART on DNA methylation have been directly tested in cultured human embryos with ART-associated aberrant methylation found at imprinting regions, including H19/IGF2[47]. No ART-associated DNA methylation change at imprinted regions was detected in neonatal blood spots in our analysis, suggesting that either the original studies reporting imprinting changes were underpowered, or that this DNA methylation signature is lost by birth. Other studies have similarly focused on imprinting regions, as well as specific genes, or genome-wide analyses in samples collected at birth or in childhood, revealing differential methylation in human and mouse placenta[48] and neonatal blood[37,49–51]. Such studies suggest that ART-induced epigenetic variation may be stable throughout pregnancy[52] and potentially even childhood[41]. It is important to note that, although we found no evidence for a specific effect of ART conception on imprinting associated DNA methylation, or on a commonly used proxy measure of global DNA methylation, our analysis cannot conclusively exclude that such effects may be revealed using alternative approaches.

The potential for ART to affect 'global' methylation has also been assessed. This is generally tested using proxy measures such as repeat-based methylation at LINE1 and/or Alu elements, or average methylation across thousands of individual probes within these elements. Using the latter approach, we found no evidence of a difference in 'global' DNA methylation in blood of ART and non-ART conception groups at birth or adulthood, despite clear evidence of an age effect (increasing with age). Our findings are in keeping with a previous analysis of first trimester chorionic villous tissue, where no evidence of an effect of ART on global methylation was found[53]. However, others have reported variation in LINE1 and/or Alu elements in blood[54] and/or placenta of ART offspring[32,55], with ART generally associated with lower methylation at these regions. Other comparable studies found no differences in LINE1 methylation in either tissue[56].

Few studies to date have implicated different aspects of ART procedures, rather than underlying infertility, in inducing epigenetic variation in the embryo, including ovarian stimulation[57] and embryo culturing[58]. For example, the specific use of ICSI has been linked to a higher SNRPN gene methylation relative to spontaneous or IVF conception[44]. Specific components within culture media may contribute to altered epigenetic status[59]. A recent study of genome wide methylation of placentas from pregnancies conceived with IVF/ICSI showed distinct epigenetic profiles relative to those conceived with less invasive procedures (ovulation induction, intrauterine insemination)[60].

The specific DMR within CHRNE that appears sensitive to ART-procedures has recently been demonstrated to show parent of origin allele-specific differential methylation. Specifically, analysis of 250 adult blood methylomes and more than 1100 transcriptomes identified significantly higher methylation on the maternally inherited allele with evidence for inter-individual variation associated with a specific genetic variant (methylation quantitative trait loci)[61]. Furthermore, the nearby age-associated DMR2 we identified is consistent with previous data in children (aged zero to 5 years) that showed a loss of methylation at this region in infancy[62]. In combination, these data suggest a complex interplay of age, genetic, sex specific and environment in regulating CHRNE gene methylation, the functional consequences of which will require further investigation.

An interesting observation is that the CHRNE DMR at birth in both cohorts, was present in both IVF (with embryo culturing) and those who underwent IUI and GIFT procedures in the absence of culturing. This implies an effect of the ovarian stimulation or subfertility itself, rather than any of the additional embryo culturing processes associated with IVF. A similar direct effects of ovarian stimulation on offspring epigenetic profile have been reported for maternally imprinted regions[63,64]. and for LINE1 methylation, which was decreased in association with high-dose hormone treatment[65].

Mounting evidence links epigenetic variation, primarily differential DNA methylation, to a range of human phenotypes and health conditions[66]. Despite this, the relevance of the ART-associated methylation variation at birth described here remains unclear, particularly as we recently reported no evidence of adverse health outcomes in the same population of ART conceived individuals following extensive phenotypic examination in adulthood[36].

There are several unique strengths of the current study. These include (i) the longitudinal analysis of blood collected soon after birth and in adulthood, (ii) the relatively large sample size compared with previous similar studies, (iii) the availability of information about the type of ART procedure employed, that allowed the effects of ovarian stimulation and culturing to be assessed and, (iv) the independent replication of ART-associated epigenetic variation in a previously published cohort. Limitations include (i) the lack of any functional assessment of the impact of the small ART-associated DNA methylation variation (5–12.9% difference) on gene expression, (ii) an inability to directly assess the effects of ovarian stimulation as a contributor to the identified epigenetic variants, (iii) limited data on other pregnancy and postnatal exposures that may affect DNA methylation of ART (e.g., medications, smoking and alcohol consumption), (iv) the use of de-convoluted whole blood for analysis does not allow us to make any comment about changes in specific blood cell types, (v) the possibility that adult participants (ART and non-ART) were self-selected as a comparatively healthy cohort and therefore less resolution of DNA methylation may be more evident in a less healthy cohort, and (vi) the number of samples, while being the largest of its type, is small relative to contemporary EWAS studies in other fields, which limits our power to detect associations between DNA methylation and specific ART procedures.

In summary, ART conception is associated with limited epigenetic variation at birth that largely attenuates by adulthood. The epigenetic variation may be associated in part with ovarian stimulation, or infertility per se. Additional studies of larger

sample size in both animal models and humans are required in order to replicate our findings. Even if the transient epigenetic changes associated with ART are replicated, the potential health implications should not be over-interpreted given the absence of any direct evidence for downstream functional consequences of the observed epigenetic change, and the lack of compelling evidence for altered health outcomes in adulthood.

## Methods

**Participants**. The protocol and details of measurements in the 'Clinical review of the Health of adults conceived following Assisted Reproductive Technologies' (CHART) study have been published previously[67]. A total of 193 ART-conceived and 86 non-ART-conceived adults gave informed consent to a have a detailed phenotypic analysis, including providing a venous blood sample and researcher access to previously collected neonatal blood spots for DNA isolation and epigenetic analysis[35,68]. Amongst the ART participants there were 147 IVF, 43 GIFT, and 3 with an unknown type of ART[36]. Matched data for both time points were available for 131 ART and 55 non-ART individuals. The study was approved by The Royal Children's Hospital Human Research Ethics Committee (RCH HREC Project 33163).

**Blood collection**. Up to 9 mL of peripheral whole blood was collected from ART and non-ART adults in Sarstedt EDTA tubes. Blood tubes were spun at $500 \times g$ for 10 min at 20 °C with no brake and full acceleration, and six 0.5 mL plasma aliquots were taken for storage. The remaining buffy coat layer was collected and mixed with Fetal Bovine Serum (FBS), 10% Dimethyl Sulfoxide (DMSO) and EDTA. The samples were aliquoted in a volume of 500 μL into barcoded cryotubes. Tubes were frozen at a controlled rate (decrease of 1 °C/min) and stored in the vapor phase of a liquid nitrogen tank until thawed for DNA extraction.

**Guthrie spot retrieval**. Neonatal blood spots (Guthrie spots) were prepared between 48 to 72 h post birth with parental informed consent and stored at room temperature. HREC approval (RCH HREC Project 33163) was obtained and informed consent from participants was sought to retrieve the Guthrie spots from New Born Screening (NBS) at Victorian Clinical Genetics Services (VCGS) of Murdoch Children's Research Institute (MCRI). Approximately nine 3 mm punches were obtained from each card.

**DNA isolation and quality control**. Whole 3 mm diameter Guthrie spots were lysed with proteinase K (Bioline Cat. No. BIO-37037) overnight then macerated with beads using the Qiagen TissueLyser II at frequency 30 for 40 s to separate blood from filter paper[69]. DNA was extracted using the Zymo Research ZR DNA-Card Extraction Kit (Cat. No. D6040), according to the manufacturer's protocol. Adult buffy coats were lysed with proteinase K for 2 h and the DNA was extracted using the Qiagen QIAamp® DNA Mini spin kit (Ref 56304). DNA was quantified using Nanodrop and quality was checked using gel electrophoresis[69].

**DNA methylation profiling**. Genomic DNA (200 to 500 ng) from Guthrie spots and whole adult blood were randomized into 96-well plates and sent to GenomeScan (Netherlands) for sodium bisulfite treatment and genome-wide methylation analysis using Illumina InfiniumMethylationEPIC BeadChips (referred to from now as 'EPIC array')[70]. The EPIC array measures DNA methylation level at more than 850,000 CpG sites (referred to as 'EPIC probes'), and covers all gene promoters, gene bodies and ENCODE-assigned distal regulatory elements[71]. Raw IDAT files were received on a hard-disk from Service XS and used for data analysis.

**Data cleaning, normalization, and statistical analysis**. Raw IDAT files were processed and analyzed using the MissMethyl and minfi packages for R[72,73], both available from Bioconductor[74]. Samples were checked for quality and those with a mean detection $p$-value of >0.01 were removed (5 neonatal and 4 adult samples), leaving 207 neonatal blood spots ($n = 149$ ART, $n = 58$ non-ART) and 233 whole adult blood ($n = 158$ ART, $n = 75$ non-ART) samples for analysis. Data were normalized for both within and between array technical variation using SWAN (Subset-quantile Within Array Normalization)[75]. Probes with poor average quality scores (detection $p$-value > 0.01), those associated with SNPs (MAF > 0%) and cross-reactive probes[71] were removed from further analysis. This left a total of 724,897 probes for neonatal blood spot analysis and 766,247 probes for adult blood analysis, of which 722,301 probes were common to both datasets. Cell composition was determined using the estimateCellCounts tool, with the 'CordBlood' reference data used for neonatal blood spot analysis[76] and the 'Blood' reference data used for adult peripheral blood analysis[77]. Differential methylation analysis by linear regression modeling was performed using limma[78]. Confounders and covariates were identified using principal component analysis (shown in Supplementary Fig. 1) and were incorporated in the analysis models as required. The final model incorporated the following covariates: Sex, EPIC array position (plate well and chip position) and cell composition proportions (CD8 T cells, CD4 T cells, B cells, Monocytes, Eosonophils, Neutrophils). Differentially methylated probes (DMPs) were those that showed an adjusted $p$-value of <0.05 (Benjamini Hochberg) and a change in methylation (delta beta or $\Delta\beta$) of ≥5%. Differentially methylated regions (DMRs) were identified using the DMRcate tool[79] and Bedtools were used to intersect DMRs with individual probes[80]. DMPs were assigned to the nearest gene within 1 megabase (1 Mb) using the GREAT tool[81]. DNA methylation at repetitive elements was calculated using the REMP (Repetitive Element Methylation Prediction) package[43].

**Reporting summary**. Further information on research design is available in the Nature Research Reporting Summary linked to this article.

## Data availability
The data sets generated and analyzed for the current study are deposited in the Gene Expression Omnibus repository with the accession number GSE131433.

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

## Acknowledgements

The authors acknowledge the participants who generously gave their time to the study. This work was made possible through the Victorian State Government Operational Infrastructure Support and the Australian Government NHMRC IRIISS. This study was funded by a National Health and Medical Research Council Project Grant (APP1099641; 2016–2017), The Royal Children's Hospital Research Foundation, Monash IVF Research and Education Foundation, and Reproductive Biology Unit Sperm Fund, Melbourne IVF. B.N. is supported by an NHMRC (Australia) CJ Martin Fellowship (1072966). DPB is supported by an NHMRC Senior Research Fellowship (1064629).

## Author contributions

R.S. and J.H. conceived and designed this study. B.N. performed differential DNA methylation analysis. B.N. and A.S. performed quality control of DNA methylation data. S.L. and J.K. collected and stored biological samples from the CHART cohort. A.C. and B.K. performed molecular biology sample processing. J.H., D.P.B., M.J., K.H., D.J.A., L.W.D., S.R., L.W., M.C., J.M. and R.M. were involved in subject recruitment, data collection, and clinical phenotyping of the CHART cohort. R.S. supervised the study. B.N. and R.S. drafted the paper, which was reviewed by all the authors.

## Additional information

**Competing interests:** The authors declare no competing interests.

