## [Peer Review File · Nature Communications]

Reviewers' comments:

Reviewer #1 (Remarks to the Author):

Using a unique longitudinal cohort of ART-conceived individuals, previously shown to have no differences in health outcomes in early adulthood compared with non-ART-conceived individuals, the authors profiled DNA methylation in blood collected in the newborn period from Guthrie Cards and in adulthood (age 22-35 years). They sought specific ART-associated variation in methylation and found significant changes, most of which occurred independently of embryo culturing. There was less evidence to support persistence of ART-associated differential methylation into adulthood. They concluded that they found specific ART-associated epigenetic variation in two independent cohorts, suggesting that infertility per se, or aspects of the technology such as ovarian stimulation, may affect the early embryo epigenome. That ART-associated epigenetic variation at birth largely resolved by adulthood with no direct evidence that it impacted on development and health.

1. Strengths The authors have sought an answer to an important question with a state-of-the-art epigenome array in a unique well-described cohort followed for many years.

2. Weaknesses The cohort is comparatively small, the blood samples are whole blood so blood counts have to be deconvoluted which limits the power of the analysis, and the array does not capture more than DNA methylation.

3. The authors have identified an interesting topic that is the potential adverse effects on long-term health associated with ART, potentially attributable to epigenetic variation induced in the periconceptual period. They have therefore established a cohort of singleton ART-conceived young adults (aged 18-28 years) and a matched non-ART group from the same source population, and using a telephone interview found an increased rate of maternally-reported hospital admissions in the former, atopic respiratory conditions and metabolic/endocrine/nutritional disease. In a follow-up they assessed vascular, cardiometabolic, anthropometric and respiratory health clinically in a subset of the original cohort, now aged 22-35 years, and found no evidence of adverse health outcomes associated with ART conception. Why was there a discrepancy between the two studies on the same cohort?

4. Using that cohort they have performed a retrospective study of longitudinal data for Epigenome-wide Association Study (EWAS) of these ART and non-ART-conceived individuals from the neonatal period through to adulthood, spanning up to 35 years since birth. The methylation difference $\Delta\beta$ of the 2340 DMPs between the groups was 0.026 ± 0.013 (with a substantial largest effect of 0.129 (i.e. 12.9%)). DMPs are difficult to identify using whole blood so a power analysis would be valuable and a priori definition of a significant DMP ($>5\%$ seems reasonable but later they start 'fishing' with lower levels).

5. The majority of DMPs (79.1%), showed a higher DNA methylation level among ART offspring in neonatal blood compared with non-ART offspring. But given the limitations of their method should they not focus on the 31 probes with $>5\%$ $\Delta\beta$ effect?

6. They then start 'fishing' as with 'Despite not reaching significance following FDR correction in adulthood, six of these 136 DMPs were also differentially methylated by $\geq 5\%$ in adulthood'. What are we to make of the sentence that follows other than they have insufficient power "Of the 136 DMPs showing greater than 5% difference between groups, all but one were within 1Mb of a gene transcription start site and 4 genes had 2 DMPs in their vicinity'.

7. They then note that 'Of 31 probes, 20 also showed a $\Delta\beta \geq 5\%$ in neonatal blood' implying that the changes persist. But they have concluded that 'There was less evidence to support persistence of ART-associated differential methylation into adulthood'. I have missed something here.

8. The search that seems on former ground is when they look for 'differentially methylated regions (DMRs), which contain multiple DMPs that show correlative methylation. DMRs, defined as

a region containing ≥ 3 DMPs, at least one of which have a $\Delta\beta \geq 5\%$.

9. Of 18 DMR (comprising 106 total probes) were identified in neonatal blood. Three DMRs common to both time-points and found near the genes *CHRNE* (7 probes), *PRSS16* (3 probes) and *TMEM18* with the same direction and similar level of DNA methylation change. Ideally they could replicate these results and they have done by finding the three strongest DMRs also covered by probes in the 450K dataset, which revealed a replication of the *CHRNE* and *PRSS16* ART-associated DMRs, both in terms of direction of effect and magnitude of difference. They then separated the cohort into three groups using different formats including IVF and IUI and they found that changes in methylation at the *CHRNE* and *PRSS16* DMRs were in all ART sub-types. That is interesting though the limitations of the 450K array to replicate EPIC should be noted.

10. They then carried out a focused analysis of 706 EPIC array probes that are located within 50 DMRs previously identified as being associated with imprinting. In the original EWAS, only 2 imprinting associated probes showed evidence of ART-associated differential methylation (at birth), Whereas approximately 9% of all 722,000 probes showed some evidence of differential methylation in neonatal blood using a more relaxed threshold, only 4% of imprinting-associated probes fell into this category. When the combined mean of these probes was compared across ART and non-ART groups, there was clear evidence of a gain in methylation with age. Though the lack of evidence does not exclude an effect given the limitations of their samples.

11. The conclusion is reasonable then is that there is evidence for specific ART-associated methylation variation around birth, some replicated in an independent cohort. The replicated lower methylation in *CHRNE* at birth implies an effect of the ovarian stimulation or subfertility itself.

12. In discussing limitations they should emphasise power and that they used whole blood and deconvoluted data according to the content of the whole blood in terms of cell subsets plus the reduce methylation sites tested with 450K arrays. I would limit analysis of data with borderline significance which could be alluded to and put in Supplementary tables.

13. Discussion is long with much speculation as to what the future might show and this could be substantially curtailed.

Professor Richard David Leslie

Reviewer #2 (Remarks to the Author):

The authors take aim at an especially controversial dimension of human ARTs: with concerns raised over the possible short and long term consequences of ART-associated epigenetic changes, what evidence can be brought to bear on the persistence of such changes into adulthood?

The analysis reported here shows indeed that certain ARTs may be associated with changes in methylation as detected in new born blood cells but that overall, these changes did not persist into adulthood for the unique cohort of patients who had been conceived by an ART. While this take home message offers some degree of comfort to the ART industry and its much heralded future growth, the design of the study and its implementation should be scrutinized until more compelling and insightful data sets become available.

This reviewer's major concerns are:

Blood good choice for entry into the study but imagine the differences in proliferation between bone marrow derived cells and the far more likely derivatives of the embryonic germ layers like neuro ectoderm, mesoderm, germline. Question is how would relative turnover histories of lineages be influenced by methylation marks imposed embryonically and at birth compared to adult outcomes?

Also, authors speculate appropriately that the only common denominator here is that of embryo culture over the years of ARTs that have been drawn upon would be embryo culture. Further, they admit that many other "ARTs" have come into and left or persisted in the field with respect to treatment strategies-like COH. Fact is, ARTs have been a moving target over the age span from

which these patients experienced their treatments and some would argue that even COH has been administered by the REI profession in anything but a consistent way (from clomid to pergonanl/HMG to recFSH/LH to rechCG, to GnRh agoists/antagonists). Caution needs to be exercised with such a possibly high impact statement coning from this design and patient cohort with special reference to the issue of how embryo culture may or may not influence methylation "then" versus much later in life.

The ART field right now is an absolute mess as both the efficacy and safety of extended culture have been driven by commercial incentives at the expense of asking the right biological questions about media, time in culture, impact of cryopreservation, biopsy etc. Clearlt the face of human ARTs continues to change and while findings like this offer some signs of encouragement, the down side of limited studies will be costly to patients and will be ripe vor exploitation by the media.

Reviewer #3 (Remarks to the Author):

Thank you for asking me to review this paper which describes an analysis of DNA methylation in individuals born following the use of assisted reproductive technologies. This is an important paper for a number of reasons - firstly there is much interest in the role of epigenetic modification in the development of disease; secondly the use of ART is increasing and evidence in support of its safety is therefore crucial and thirdly there are relatively few longitudinal studies in this area. The authors show that ART is associated with limited epigenetic variation at birth and virtually none in adulthood. The findings are of interest to those in a number of fields and additionally to the public.

The study appears to have been well done and the paper is well-written and easy to read.

I have a few comments which the authors could address in a revision:

1. In the introduction (line 91) the authors state that adverse effects of ART on health 'are likely attributable to epigenetic variation induced...'. I think this is a bit strong, although there are many studies which show there may be epigenetic differences, there is virtually no evidence which confirms causation.

2. The sample sizes in this study and the confirmatory cohort are small, but I imagine there are simply no larger longitudinal cohorts. Could the authors include a discussion on sample size in the strengths and limitations section?

3. The changes in DNA methylation identified in this, the confirmatory cohort and many of the other studies are very small. Thus, their biological significance is unclear. The authors do mention that they were unable to obtain expression data but I would like to see some discussion of the small changes in methylation and likelihood of them being important.

4. Line 288 - suggest change 'differential methylation in infancy' to 'in neonates'

5. Could the authors comment on the function of the genes they have identified? I note that CHRNE is of developmental importance at about the time of birth and I wonder if differential methylation at that time might just represent an altered trajectory of maturation in the ART neonates? Were they more likely to be exposed to glucocorticoids which might accelerate maturation for example?

Response to Reviewer comments

Reviewer #1:

Using a unique longitudinal cohort of ART-conceived individuals, previously shown to have no differences in health outcomes in early adulthood compared with non-ART-conceived individuals, the authors profiled DNA methylation in blood collected in the newborn period from Guthrie Cards and in adulthood (age 22-35 years). They sought specific ART-associated variation in methylation and found significant changes, most of which occurred independently of embryo culturing. There was less evidence to support persistence of ART-associated differential methylation into adulthood.

They concluded that they found specific ART-associated epigenetic variation in two independent cohorts, suggesting that infertility per se, or aspects of the technology such as ovarian stimulation, may affect the early embryo epigenome. That ART-associated epigenetic variation at birth largely resolved by adulthood with no direct evidence that it impacted on development and health.

1. Strengths The authors have sought an answer to an important question with a state-of-the-art epigenome array in a unique well-described cohort followed for many years.

RESPONSE: We are pleased the reviewer acknowledges the importance of the question, the uniqueness of our cohort, and the state of the approach for assessing epigenetic variation

2. Weaknesses The cohort is comparatively small, the blood samples are whole blood so blood counts have to be deconvoluted which limits the power of the analysis, and the array does not capture more than DNA methylation

RESPONSE: Our longitudinal ART cohort, spanning the early postnatal period to adulthood is unique internationally due to the scarcity of studies with ART conceived adults worldwide. This is the primary reason for our relatively modest sample size relative to other contemporary (mostly cross sectional studies). In order to address this and provide additional confidence in the findings we have applied stringent significance cut-offs for variable methylation within our study AND have sought to replicate positive associations in a completely independent cohort.

We agree that the use of genomic DNA from whole blood does not allow for the mapping of epigenetic variation within specific cell populations, or the mapping of epigenetic processes beyond DNA methylation. These limitations are common to most contemporary EWAS studies but are both very interesting questions to explore in modern longitudinal cohorts. We aim to explore these questions in cohorts currently in development in several settings internationally.

3. The authors have identified an interesting topic that is the potential adverse effects on long-term health associated with ART, potentially attributable to epigenetic variation induced in the periconceptual period. They have therefore established a cohort of singleton ART-conceived young adults (aged 18-28 years) and a matched non-ART group from the same source population, and using a telephone interview found an increased rate of maternally-reported hospital admissions in the former, atopic respiratory conditions and metabolic/endocrine/nutritional disease. In a follow-up they assessed vascular, cardiometabolic, anthropometric and respiratory health clinically in a subset of the original cohort, now aged 22-35 years, and found no evidence of adverse health outcomes associated with ART conception.

Why was there a discrepancy between the two studies on the same cohort?

RESPONSE: In terms of adverse health outcomes, the 1st study collected data using a telephone interview with the mother. The increased rate of hospitalizations in the ART group were in the 1st 18 years of life and are detailed in the original paper e.g. hernia repairs 3.8% vs 2.2%; genitourinary 4.8% vs 2.3%; dental surgery 6.2% vs 3.3%; tonsillitis/tonsillectomy 12.3% vs 4.3%. Some of this was thought to be attributed to parental vigilance associated with perceived vulnerability in the much

wanted babies – especially in those pioneering years for ART. We did not ask the young adults themselves about hospitalizations in the 2nd study when they were older.

Respiratory conditions identified as potentially problematic in the 1st interview-based study were checked in the 2nd study by doing clinical assessments (forced expiratory airflow rates). No differences were found between ART and non-ART, having adjusted analyses for perinatal and other important confounders. Although ART participants reported having had some “asthma, lung or breathing problems”, these had mostly resolved by adulthood and ongoing rates were similar in the ART and non-ART groups.

The third area of difference between ART and non-ART participants in the 1st study was of borderline significance, encompassing a heterogeneous ICD-10 category, metabolic/endocrine/nutritional disease. Overall, these were uncommon, present in 3.3% of ART and 2.2% of non-ART. This category included small numbers with Type 1 diabetes (5 ART vs 3 non-ART), polycystic ovary syndrome and lactose intolerance. We do in fact know that some ART and non-ART individuals with these conditions did not participate in the 2nd study and will add this as a limitation to the study, i.e. the possibility that there has been a healthy participant bias (in both groups). Importantly, however, less than 30% of participants in the 2nd study reported their “physical health in general” as excellent.

4. Using that cohort they have performed a retrospective study of longitudinal data for Epigenome-wide Association Study (EWAS) of these ART and non-ART-conceived individuals from the neonatal period through to adulthood, spanning up to 35 years since birth. The methylation difference delta beta of the 2340 DMPs between the groups was 0.026 ± 0.013 (with a substantial largest effect of 0.129 (i.e. 12.9%)). DMPs are difficult to identify using whole blood so a power analysis would be valuable and a priori definition of a significant DMP (>5% seems reasonable but later they start ‘fishing’ with lower levels).

RESPONSE: We initially report the full number of 2,340 DMPs that show an association with ART in neonates following adjustment for multiple testing ($p_{adj} < 0.05$; page 8). We then further explored a subset of the most differentially methylated 136 DMPs (Fig 1B) from this list with a $p_{adj} < 0.05$ and a change of methylation $\Delta B > 0.05$ in order to gain a more detailed understanding of the genomic regions most sensitive to ART conception.

As there were no probes that survived the adjusted p-value cutoff in adults, we performed an exploratory analysis of the top ranked probes in the list (cross referencing to infancy to look for commonalities at both ages), which may be confusing to some readers. To address this, we have deleted Supplementary Figure 2. Therefore Supplementary Figure 3 becomes Supplementary Figure 2, and so on. Figure legends and in text references to the supplementary figures have been altered accordingly.

Despite this, we still see the value in providing these data for readers and have therefore retained Supplementary Table 2, but have deleted the following text:

‘Although not reaching significance after correction for multiple testing, 31 probes showed a methylation difference of delta beta of ≥ 0.05 (5% methylation change) and non-adjusted p-value < 0.01 between ART and non-ART in adult blood (Supplementary Figure 2; Supplementary Table 2). Of these 31 probes, 20 also showed a delta beta of ≥ 0.05 (5% methylation change) in neonatal blood (Supplementary Figure 2).’

5. The majority of DMPs (79.1%), showed a higher DNA methylation level among ART offspring in neonatal

blood compared with non-ART offspring. But given the limitations of their method should they not focus on the 31 probes with >5% delta effect?

RESPONSE: Again, there may have been some confusion due to us reporting data for adult DMPs that do not reach significance following correction for multiple testing (31 probes in Supplementary Figure 2B and C). We removed these from the text and supplementary figure (please see answer to point 4 above). In fact, we now focus on 136 probes **in neonates** that remain significantly associated with ART procedures following correction for multiple testing (116 hyper and 21 hypo-methylated that show a delta beta difference of >0.05 (5% methylation) (Figure 1,3,4) and then map DMRs in the vicinity of these probes (Figure 2-4).

6. They then start 'fishing' as with 'Despite not reaching significance following FDR correction in adulthood, six of these 136 DMPs were also differentially methylated by $\geq 5\%$ in adulthood'. What are we to make of the sentence that follows other than they have insufficient power "Of the 136 DMPs showing greater than 5% difference between groups, all but one were within 1Mb of a gene transcription start site and 4 genes had 2 DMPs in their vicinity".

RESPONSE: In this section (Figure 1C and D) we simply plotted the change in DNA methylation at p-adjusted DMPs in neonates and adults. We think that this is a powerful way to show that the 136 significant differences around birth largely disappear by adulthood. In order to discuss the findings, it is important to highlight that 6 of these DMPs show some evidence of a persistent difference in methylation between groups from birth to adulthood. The sentence that follows "all but one were within 1Mb of a gene" is purely to describe the location of the p-adjusted significant probes relative to genes and does not deal with strength of association with ART following statistical analysis. We believe this misunderstanding is related to the point below (7.)

7. They then note that 'Of 31 probes, 20 also showed a delta/beta $\geq 5\%$ in neonatal blood' implying that the changes persist. But they have concluded that 'There was less evidence to support persistence of ART-associated differential methylation into adulthood'. I have missed something here.

RESPONSE: Please see answer to point 4 and 6 above.

8. The search that seems on firmer ground is when they look for 'differentially methylated regions (DMRs), which contain multiple DMPs that show correlative methylation. DMRs, defined as a region containing ≥ 3 DMPs, at least one of which have a delta/beta $\geq 5\%$ '.

RESPONSE: Although it is correct that identifying differentially methylated regions (DMRs) is an important step towards defining the full scope of DNA Methylation variation, it is equally important to identify DMPs. The majority of the literature in the field uses a combined approach as we have here.

9. Of 18 DMR (comprising 106 total probes) were identified in neonatal blood. Three DMRs common to both time-points and found near the genes *CHRNE* (7 probes), *PRSS16* (3 probes) and *TMEM18* with the same direction and similar level of DNA methylation change. Ideally they could replicate these results and they have done by finding the three strongest DMRs also covered by probes in the 450K dataset, which revealed a replication of the *CHRNE* and *PRSS16* ART-associated DMRs, both in terms of direction of effect and magnitude of difference. They then separated the cohort into three groups using different formats including IVF and IUI and they found that changes in methylation at the *CHRNE* and *PRSS16* DMRs were in all ART sub-types. That is interesting though the limitations of the 450K array to replicate EPIC should be noted.

RESPONSE: We are pleased that the reviewer has identified the significance of our replication of key findings in an independent cohort (albeit a less state of the art platform). In order to highlight this we have included the following:

“The disadvantage of using the 450K array for validation is that it does not include all EPIC probes and therefore generates a lower resolution picture of DNA methylation profile. Despite this, there is a strong correlation between methylation values generated by the 450K and EPIC arrays, which allows us to confirm a subset of our findings (Solomon Epigenetics 2018, PMID: 30044683).

10. They then carried out a focused analysis of 706 EPIC array probes that are located within 50 DMRs previously identified as being associated with imprinting. In the original EWAS, only 2 imprinting associated probes showed evidence of ART-associated differential methylation (at birth), Whereas approximately 9% of all 722,000 probes showed some evidence of differential methylation in neonatal blood using a more relaxed threshold, only 4% of imprinting-associated probes fell into this category. When the combined mean of these probes was compared across ART and non-ART groups, there was clear evidence of a gain in methylation with age. Though the lack of evidence does not exclude an effect given the limitations of their samples.

RESPONSE: We agree that we certainly cannot rule out an effect at imprinted regions given the limitation of our samples. However, it is important to note that using this high resolution approach, we found no evidence for a systematic effect of ART conception on imprinted regions relative to the genome as a whole. We added the following sentence to this section of the paper (page 13, last paragraph):

“It is important to note that, although we found no evidence for a specific effect of ART conception on imprinting associated DNA methylation, or on a commonly used proxy measure of global DNA methylation, our analysis cannot conclusively exclude that such effects may be revealed using alternative approaches.”

11. The conclusion is reasonable then is that there is evidence for specific ART- associated methylation variation around birth, some replicated in an independent cohort. The replicated lower methylation in CHRNE at birth implies an effect of the ovarian stimulation or subfertility itself.

RESPONSE: We are pleased that the reviewer agrees with our conclusion.

12. In discussing limitations they should emphasise power and that they used whole blood and deconvoluted data according to the content of the whole blood in terms of cell subsets plus the reduce methylation sites tested with 450K arrays. I would limit analysis of data with borderline significance which could be alluded to and put in Supplementary tables.

RESPONSE: We agree with the reviewer about reporting borderline results. For this reason we removed Adult-DMPs in Supplementary Figure 2C, and instead only show them in Supplementary Table 2. These Adult-DMPs had an uncorrected $p < 0.01$, but did not survive multiple-testing correction.

Regarding limitations of the study, we have included the following sentences in the strengths and limitations:

“..., (iv) the use of de-convoluted whole blood for analysis does not allow us to make any comment about changes in specific blood cell types and (iv) the number of samples, while being the largest of its type, is small relative to contemporary EWAS studies in other fields, which limits our power to detect associations between DNA methylation and specific ART procedures.”

13. Discussion is long with much speculation as to what the future might show and this could be

substantially curtailed.

RESPONSE: We accept that the discussion is long and have deleted some of the more speculative text on page 14 and 15.

Reviewer #2 (Remarks to the Author):

1. The authors take aim at an especially controversial dimension of human ARTs: with concerns raised over the possible short and long term consequences of ART-associated epigenetic changes, what evidence can be brought to bear on the persistence of such changes into adulthood? The analysis reported here shows indeed that certain ARTs may be associated with changes in methylation as detected in new born blood cells but that overall, these changes did not persist into adulthood for the unique cohort of patients who had been conceived by an ART. While this take home message offers some degree of comfort to the ART industry and its much heralded future growth, the design of the study and its implementation should be scrutinized until more compelling and insightful data sets become available.

RESPONSE: We agree that our study is far from definitive and that further analysis is warranted – both in the context of increasing sample size and in exploring the epigenetic legacy of ART in different cell types. This is outside the scope of our current approach due to the constraints imposed on both sample type and size by our longitudinal study design

This reviewer's major concerns are:

2. Blood good choice for entry into the study but imagine the differences in proliferation between bone marrow derived cells and the far more likely derivatives of the embryonic germ layers like neuro ectoderm, mesoderm, germline. Question is how would relative turnover histories of lineages be influenced by methylation marks imposed embryonically and at birth compared to adult outcomes?

RESPONSE: These are valid and important questions that can be addressed in subsequent studies in the field. Note constraints on our sample discussed in the previous point.

3. Also, authors speculate appropriately that the only common denominator here is that of embryo culture over the years of ARTs that have been drawn upon would be embryo culture. Further, they admit that many other "ARTs" have come into and left or persisted in the field with respect to treatment strategies-like COH. Fact is, ARTs have been a moving target over the age span from which these patients experienced their treatments and some would argue that even COH has been administered by the REI profession in anything but a consistent way (from clomid to pergonanl/HMG to recFSH/LH to rechCG, to GnRh agoists/antagonists). Caution needs to be exercised with such a possibly high impact statement coning from this design and patient cohort with special reference to the issue of how embryo culture may or may not influence methylation "then" versus much later in life.

RESPONSE: We presume the reviewer is referring to authors of other publications as we do not make mention of this. We in fact have been able to compare culture (IVF) and no culture (GIFT) conditions and find no significant differences between these 2 groups (Supp Fig 6D, in Results text on page 7 and repeated in Discussion on pages 9/10). Higher up on page 7, in relation to DMR results in all ART subtypes, we state that "ART-associated effects are not associated with embryo culture specifically, but with other steps in the ART process or potentially infertility in general." We do not know which statement Reviewer 2 is referring to that is "possibly high impact" in relation to embryo culture. By highlighting the results and accompanying text in the above paragraph, we hope embryo culture concerns have been addressed.

We agree that it is very important to emphasise the limitations of this study and the need for this extremely important research to be repeated in other cohorts.

4. The ART field right now is an absolute mess as both the efficacy and safety of extended culture have been driven by commercial incentives at the expense of asking the right biological questions about media, time in culture, impact of cryopreservation, biopsy etc. Clearly the face of human ARTs continues to change and while findings like this offer some signs of encouragement, the down side of limited studies will be costly to patients and will be ripe for exploitation by the media.

RESPONSE: We agree that a degree of caution is warranted when reporting findings in this field, which is why we have taken considerable care not to overstate findings and to report only robust DNA methylation associations that are within our power to detect. Despite this it is important that findings in the field be published so that they can be directly compared and contrasted across different ages and settings. We expect that future analyses will combine our dataset with other cohorts to specifically look at different aspects of ART procedures.

Reviewer #3 (Remarks to the Author):

Thank you for asking me to review this paper which describes an analysis of DNA methylation in individuals born following the use of assisted reproductive technologies. This is an important paper for a number of reasons - firstly there is much interest in the role of epigenetic modification in the development of disease; secondly the use of ART is increasing and evidence in support of its safety is therefore crucial and thirdly there are relatively few longitudinal studies in this area. The authors show that ART is associated with limited epigenetic variation at birth and virtually none in adulthood. The findings are of interest to those in a number of fields and additionally to the public. The study appears to have been well done and the paper is well-written and easy to read.

RESPONSE: We are pleased about the reviewer's positive assessment of (1) the importance and timeliness of our study, (2) the uniqueness of our dataset and (3) the approach we took for analysis and presentation of our data.

I have a few comments which the authors could address in a revision:

1. In the introduction (line 91) the authors state that adverse effects of ART on health 'are likely attributable to epigenetic variation induced...'. I think this is a bit strong, although there are many studies which show there may be epigenetic differences, there is virtually no evidence which confirms causation.

RESPONSE: We have changed the sentence from: 'likely attributable' to 'some of which may be attributable'

2. The sample sizes in this study and the confirmatory cohort are small, but I imagine there are simply no larger longitudinal cohorts. Could the authors include a discussion on sample size in the strengths and limitations section?

RESPONSE: We agree and this has been addressed as per Reviewer 1 comment 12.

3. The changes in DNA methylation identified in this, the confirmatory cohort and many of the other studies are very small. Thus, their biological significance is unclear. The authors do mention that they were unable to obtain expression data but I would like to see some discussion of the small changes in methylation and likelihood of them being important.

RESPONSE: We agree that some may find the effect sizes reported to be relatively small. However, these are in fact comparatively large relative to other contemporary EWAS studies, including large meta analyses for exposure to smoking, maternal folate levels and birthweight. Nevertheless, we have now addressed this issue in strength and Limitations section and have changed the text slightly to further address the reviewer's comment:

New text:

"Limitations include (i) the lack of any functional assessment of the impact of the small ART-associated DNA methylation variation (5% - 12.9% difference) on gene expression."

4. Line 288 - suggest change 'differential methylation in infancy' to 'in neonates'

RESPONSE: done

5. Could the authors comment on the function of the genes they have identified? I note that CHRNE is of developmental importance at about the time of birth and I wonder if differential methylation at that time might just represent an altered trajectory of maturation in the ART neonates? Were they more likely to be exposed to glucocorticoids which might accelerate maturation for example?

Answer: Although we agree that this is an interesting aspect of the study, it is important that we do not over interpret the significance of the variable methylation in relation to long term ART health outcomes. As such, we have addressed this in the discussion in the following way (page 14-15):

“The specific DMR within CHRNE that appears sensitive to ART-procedures has recently been demonstrated to show parent of origin allele-specific differential methylation. Specifically, analysis of 250 adult blood methylomes and more than 1,100 transcriptomes identified significantly higher methylation on the maternally inherited allele with evidence for inter-individual variation associated with a specific genetic variant (methylation quantitative trait loci).⁷⁵ Furthermore, the nearby age-associated DMR2 we identified is consistent with previous data in children (aged zero to 5 years) that showed a loss of methylation at this region in infancy.⁷⁶ In combination, these data suggest a complex interplay of age, genetic, sex specific and environment in regulating CHRNE gene methylation, the functional consequences of which will require further investigation.”

REVIEWERS' COMMENTS:

Reviewer #1 (Remarks to the Author):

I found the responses suitable and the results are both novel and of interest to the community and the wider field of science and medicine.

I appreciate that the subject is very sensitive and I think the term on line 49 of the Abstract could be changed from

"less evidence" to "limited evidence"

they have suitably shortened the article and the Discussion and they have modified the more speculative comments.

Reviewer #3 (Remarks to the Author):

The authors have addressed the comments raised

REVIEWERS' COMMENTS:

Reviewer #1 (Remarks to the Author):

I found the responses suitable and the results are both novel and of interest to the community and the wider field of science and medicine.

RESPONSE: We are pleased that the reviewer found our responses suitable and the study novel and interesting.

I appreciate that the subject is very sensitive and I think the term on line 49 of the Abstract could be changed from

"less evidence" to "limited evidence"

RESPONSE: The abstract has been edited.

they have suitably shortened the article and the Discussion and they have modified the more speculative comments.

RESPONSE: We agree that shortening of the Discussion helped the paper and are pleased that the reviewer found the changes appropriate.

Reviewer #3 (Remarks to the Author):

The authors have addressed the comments raised

RESPONSE: We are pleased with this assessment and thank the reviewers for their valuable input.